# The Immunomodulatory Effect of Various Anaesthetic Practices in Patients Undergoing Gastric or Colon Cancer Surgery: A Systematic Review and Meta-Analysis of Randomized Clinical Trials

**DOI:** 10.3390/jcm12186027

**Published:** 2023-09-18

**Authors:** Georgios Konstantis, Georgia Tsaousi, Elisavet Kitsikidou, Dimitrios Zacharoulis, Chryssa Pourzitaki

**Affiliations:** 1Clinical Pharmacology, Faculty of Medicine, School of Health Sciences, Aristotle University of Thessaloniki, 54124 Thessaloniki, Greece; gdkonstantis@gmail.com; 2Department of Gastroenterology, Hepatology and Transplant Medicine, Medical Faculty, University of Duisburg-Essen, 45147 Essen, Germany; 3Department of Anesthesiology and ICU, Medical School, Aristotle University of Thessaloniki, 54124 Thessaloniki, Greece; tsaousig@otenet.gr; 4Department of Internal Medicine, Evangelical Hospital Dusseldorf, 40217 Dusseldorf, Germany; eliskits@gmail.com; 5Department of Surgery, University of Thessaly, Biopolis, 41110 Larissa, Greece; zacharoulis@uth.gr

**Keywords:** immunomodulation, anaesthesia, gastric cancer, colon cancer, surgery

## Abstract

Background: Gastric and colorectal carcinomas are associated with increased mortality and an increasing incidence worldwide, while surgical resection remains the primary approach for managing these conditions. Emerging evidence suggests that the immunosuppression induced by the chosen anaesthesia approach, during the perioperative period, can have a significant impact on the immune system and consequently the prognosis of these patients. Aim: This systematic review aims to comprehensively summarize the existing literature on the effects of different anaesthesia techniques on immune system responses, focusing on cellular immunity in patients undergoing the surgical removal of gastric or colorectal carcinomas. There is no meta-analysis investigating anaesthesia’s impact on immune responses in gastric and colorectal cancer surgery. Anaesthesia is a key perioperative factor, yet its significance in this area has not been thoroughly investigated. The clinical question of how the anaesthetic technique choice affects the immune system and prognosis remains unresolved. Methods: Major electronic databases were searched up to February 2023 to May 2023 for relevant randomized controlled trials (RCTs). The study protocol has been registered with Prospero (CRD42023441383). Results: Six RCTs met the selection criteria. Among these, three RCTs investigated the effects of volatile-based anaesthesia versus total intravenous anaesthesia (TIVA), while the other three RCTs compared general anaesthesia alone to the combination of general anaesthesia with epidural anaesthesia. According to our analysis, there were no significant differences between TIVA and volatile-based anaesthesia, in terms of primary and secondary endpoints. The combination of general anaesthesia with epidural analgesia had a positive impact on NK cell counts (SMD 0.61, 95% CI 0.28 to 0.94, I^2^ 0.0% at 24 and 72 h after the operation), as well as on CD4^+^ cells (SMD 0.59, CI 95% 0.26 to 0.93, I^2^ 0.0%). However, the CD3^+^ cell count, CD4^+^/CD8^+^ ratio, neutrophil-to-lymphocyte ratio (NLR), IL-6 and TNF-α levels remained unaffected. Conclusions: The combination of epidural analgesia and general anaesthesia can potentially improve, postoperatively, the NK cell count and CD4^+^ cell levels in gastric or colon surgery patients. However, the specific impact of TIVA or volatile-based anaesthesia remains uncertain. To gain a better understanding of the immunomodulatory effects of anaesthesia, in this particular group of cancer patients, further well-designed trials are required.

## 1. Introduction

Cancer remains one of the major determinants of morbidity and mortality worldwide [1]. Despite significant progress in the field of cancer prevention, diagnosis and targeted treatments, namely immunomodulatory molecules, surgical intervention constitutes the main treatment modality for the majority of cancer types.

Metastasis is a pivotal event in the progression of cancer, characterized by a complex pathophysiological process [2], primarily involving tumour angiogenesis and evasion of the host’s cell-mediated immunity [3,4]. The stress response, elicited by the surgical excision of the tumour, induces a temporary alteration of the immune system, further promoting an increase in the minimal residual disease and cancer cell burden [5]. Specifically, the reduction in the number and cytotoxic activity of T lymphocytes, characterized by CD4^+^, CD8^+^ and CD3^+^, as well as natural killer cell (NK cell) dysfunction, as well as the excessive proinflammatory responses reflected by an increased lymphocyte-to-neutrophil ratio (LNR), are pivotal characteristics of perioperative cytokine cascade activation [6]. The aforementioned immune system dysfunction hinders the elimination of cancer cells, which in turn, might lead to relapse and metastasis development.

The aspect that perioperative conditions can influence the growth of carcinomas, as well as the prognosis and tumour progression, has been proposed during the last decades [7,8]. Studies examining the effects of common anaesthetics on tumour cells in rodents have shown a correlation between the type of anaesthesia and the incidence of metastasis [9,10]. For example, propofol may possess anticancer properties, as studies in rodents have demonstrated that propofol-based anaesthesia can enhance the cytotoxicity of NK cells, reduce cancer invasiveness and promote the activation and differentiation of T lymphocytes [11,12,13]. However, there is growing evidence suggesting that certain volatile anaesthetics, such as isoflurane, may be associated with increased tumour proliferation and migration [14]. On the contrary, regional anaesthesia seems to enhance the host’s immune response against cancer cells, mainly through surgery-induced stress alleviation and NK cell function improvements [15,16].

Gastric and colorectal cancers are the most common gastrointestinal tumours, with an increasing incidence worldwide. Considering that a significant number of patients with these tumours undergo multiple surgical procedures, careful consideration should be given to the chosen anaesthesia techniques. Based on the available basic research data, it was hypothesized that perioperative immunosuppression, elicited by the implemented anaesthetic approach, could potentially exert a significant effect on the immune system and consequently the prognosis of patients undergoing surgical resection for malignant tumours. However, the existing data in this area are inconclusive, while there is no meta-analysis examining the effects of anaesthesia on patients with gastric and colorectal carcinoma. The objective of this meta-analysis is to provide clinicians with an up-to-date and comprehensive review of the best available evidence, concerning the impact of anaesthesia techniques, mainly focusing on cellular immunity inflammatory cytokines and the leukocyte-to-neutrophil ratio, in patients with gastric and colorectal cancer. The novelty of the present study lies in the fact that this is the first systematic review and meta-analysis focusing on the effect of various anaesthesia techniques on the immune system of patients undergoing surgery for gastric or colorectal cancer. Anaesthesia stands as one of the most alterable perioperative factors, and thus far, its importance in this field has not been adequately explored. Clinically, the pivotal question of whether the choice of anaesthetic affects the immune system and, consequently, prognosis remains unanswered.

## 2. Materials and Methods

The present systematic review and meta-analysis adhered to the guidelines outlined in the PRISMA statement. A systematic and comprehensive search was performed using the electronic databases PubMed Scopus and Cochrane database from February 2023 to May 2023 to identify relevant studies. The study protocol has been registered with PROSPERO under registration number CRD42023441383.

### 2.1. Eligibility Criteria

This meta-analysis included randomized clinical trials (RCTs) that investigated the impact of diverse anaesthetic techniques on cellular immunity inflammatory cytokines and the LFR in patients undergoing the resection of primary colon or gastric tumours. More specifically, the present study focused on investigating the impact of general anaesthesia using propofol, isoflurane, sevoflurane or desflurane, both as standalone methods or in conjunction with local anaesthesia. Local anaesthesia methods included regional nerve blocks, epidural anaesthesia and spinal anaesthesia. The selected studies were characterized as eligible if they assessed the outcomes of interest, comparing general anaesthesia, alone or in combination, with regional anaesthesia. Studies that did not utilize any of the aforementioned anaesthetic technique combinations were excluded from this meta-analysis. The inclusion criteria for participants were limited to adults only. No limitations were imposed on the utilized analgesia protocols. Moreover, no exclusion criteria were established regarding to the type of surgical procedure (endoscopic, open, laparoscopic, robotic), disease stage and comorbidities. Additionally, the duration of hospitalization was not taken into consideration. Language restrictions were not applied, in order to maximize the potential search results and thus minimize potential systematic errors of publication.

### 2.2. Outcome Measurements

The primary endpoints included the effect of anaesthesia techniques on cellular immunity, as measured by the CD3^+^ CD4^+^ CD8^+^ NK cell count and CD4^+^/CD8^+^ ratio, after anaesthesia induction, at the end of the surgical procedure and at 24 h and 72 h post-operation. Moreover, the secondary outcomes included the effect on the IL-6 TNF-a and LFR ratio at the same time points. An available cases approach was implemented, and participants who were excluded from the analysis were assumed to be missing at random.

### 2.3. Data Collection and Extraction

The relevant records were imported into Endnote 19 and any duplicate entries were eliminated. Two reviewers (GK and TG) independently screened the records, based on their titles and abstracts. Subsequently, eligible studies were assessed in the full text. In cases of disagreements, during the study selection process, a third reviewer (CP) resolved the discrepancies. Data including the year of publication, follow-up period, country of origin, study type, patient characteristics (gender and age), tumour type and stage, anaesthesia type and duration, surgical technique and details regarding primary and secondary endpoints, which were recorded in a predefined format.

### 2.4. Quality Assessment

The risk of bias for the primary outcomes was assessed using The Cochrane Risk of Bias (ROB) tool 2.0. [17], which evaluates five quality features of the assessed RCTs: randomization, deviations from intended interventions, missing outcomes, data reporting of results and outcome measurements. Studies were categorized as low risk if they met all the domains with a low risk of bias, high risk if they met any domain with a high risk of bias and as having some concerns if they did not fit into either category. Two independent reviewers (GK, TG) evaluated the included studies, and any discrepancies were resolved by a third reviewer (CP).

### 2.5. Statistical Analysis

Continuous data were synthesized using Hedge’s estimation method when different scales were used by the individual studies. The results were expressed as standardized mean difference (SMD) and 95% confidence intervals (CIs). For all analyses, final values were utilized, as all studies included in the analysis were randomized controlled trials. When the necessary measures of dispersion were not reported in the studies, the methods described in the Cochrane Textbook were applied [18] in order to calculate the corresponding dispersion measures. A random effects model was chosen as the preferred approach for analysis. The heterogeneity variance was estimated using the restricted maximum likelihood model. In addition, sensitivity analyses were conducted, taking into account the risk of bias in each study and the type of cancer, on the results, as well as the operative method, when a sufficient amount of data was available. Available case analysis was chosen, assuming that the missing data are due to random factors. Moreover, an available case analysis was performed, assuming that the missing data occurred randomly and were not influenced by the form of anaesthesia. All statistical analyses were conducted using STATA SE, version 16.1 (Stata Corp), utilizing the “meta” suite of commands.

## 3. Results

### 3.1. Search Results

Six studies, including 559 patients in total, met the inclusion criteria. Three RCTs [19,20,21], with 319 patients, examined the different effects of volatile-based or total intravenous (TIVA) anaesthesia, in patients undergoing laparoscopic or general operations for gastric or colorectal cancer. Three RCTs [22,23,24], including 240 patients, compared the use of general anaesthesia, alone versus general anaesthesia combined with epidural anaesthesia and analgesia. Basic characteristics of the studies and participants are summarized in Table 1. The study selection process and the reasons for exclusion are explicitly presented in Figure 1.

### 3.2. Baseline Characteristics

Three studies [19,20,21] examined the effect of TIVA using propofol administration versus sevoflurane-based anaesthesia, in patients with either colorectal [19,21] or gastric cancer [20]. In the study by Chen Y et al. [19], the laparoscopic management of colorectal carcinoma was performed exclusively, while in the study of Oh-CS et al. [21], various surgical methods were implemented, on the basis of the tumour’s location and extent. In the study by Kim YN et al. [20], the resection of the gastric carcinoma was conducted using a robotic or laparoscopic approach, and the type of gastrectomy performed was either subtotal, total or proximal subtotal, depending on the specific case and tumour location.

Three RCTs [22,23,24] investigated the comparison between general anaesthesia alone and the combination of general anaesthesia combined with epidural anaesthesia. Among the aforementioned studies, two RCTs [22,23] examined the use of inhalational anaesthesia with either isoflurane or sevoflurane and one [24] evaluated the use of TIVA. Two of the studies [23,24] focused on patients with gastric cancer, while one study [22] involved patients with colorectal cancer. The surgical techniques varied among the studies and included both laparoscopic and open techniques. In the epidural anaesthesia group, lidocaine [23,24] and either bupivacaine or fentanyl [22] were administered. The stage of the carcinoma varied across the studies from TNM stage I to stage IV. In all studies, opioids were administered intraoperatively. All relevant characteristics, pertaining to the intraoperative anaesthesia/analgesia and postoperative analgesia are presented in Table 2.

### 3.3. Risk of Bias of the Included Studies

The systematic risk of bias for all six studies was evaluated by two independent reviewers (GK, SE) using ROB 2.0. The results are presented in Table 3. Due to the limited number of relevant studies, publication bias could not be investigated. Two [20,21] out of the three studies [19,20,21] that examined the effects of propofol versus sevoflurane were assessed as having a low risk of bias, while one study [22] comparing general anaesthesia with the combination of general anaesthesia and epidural anaesthesia was found to have a high risk of bias. The remaining studies [19,23,24] were rated as having “some concerns”, primarily due to the absence of available protocols.

#### 3.3.1. Analysis of Primary Outcomes

a.NK cell counts: propofol versus sevoflurane

Two of the three studies [19,21] reported the effects of TIVA or sevoflurane-based anaesthesia on the NK cell count. Data were extracted regarding the number of NK cells after the induction of anaesthesia and 24 h postoperatively. Although the use of TIVA led to a higher concentration of NK cells, immediately after the induction of anaesthesia (SMD 0.32, CI 95% −0.04 to 0.68, I^2^ 16.09%) (Figure 2), as well as 24 h postoperatively (SMD 0.21, CI 95% −0.08 to 0.50, I^2^ 0.0%) (Figure 3), no statistical significance was achieved.

b.NK cell counts: general anaesthesia versus general anaesthesia plus epidural anaesthesia

Two of the three studies that compared general anaesthesia alone versus general anaesthesia combined with epidural anaesthesia reported data on the NK cell count. In the study conducted by Zhao J et al. [23], the NK cell count was measured immediately after the anaesthesia induction and 72 h postoperatively, while in the study conducted by Zhou Min et al. [24], measurements were taken at several time points. The meta-analysis showcased that the combination of general and regional anaesthesia significantly increased the NK cell count after the induction of anaesthesia, as well as 72 h after the operation (SMD 0.61, CI 95% 0.28 to 0.94, I^2^ 0.0% and SMD 0.49, CI 95% 0.16 to 0.82, I^2^ 0.0%) (Figure 4 and Figure 5 respectively).

c.CD3^+^, CD4^+^, CD8^+^ and C4^+^/CD8^+^ ratio: propofol versus sevoflurane

Regarding the effect of propofol and volatile anaesthesia with sevoflurane on specific lymphocyte subtypes, namely CD3^+^, CD4^+^, CD8^+^ and the CD4^+^/CD8^+^ ratio, data were obtained from two studies [19,21]. In the study conducted by Chen Yi Jiao et al. [19], patients receiving general anaesthesia with propofol were compared with those sedated with sevoflurane, and the results showed that the propofol group exhibited a higher CD8^+^ cell count postoperatively. However, these increases were attenuated 24 h after surgery. In their study, Chung-Sik et al. [21] found that patients who were sedated with sevoflurane showed a higher CD8^+^ count postoperatively and 24 h after the surgery. Data pertaining to the CD8^+^ cell count, immediately after surgery and 24 h later, were pooled, and our analysis revealed that TIVA resulted in a slight reduction during the first time point (SMD 0.08, CI 95% −0.49 to 0.65, I^2^ 55.84%) (Figure 6). On the other hand, propofol favoured a slight increase in CD8^+^ cell levels 24 h after the operation (SMD −0.10, CI 95% −0.68 to 0.49, I^2^ 58.1%) (Figure 7). In both cases, statistical significance was not observed.

d.CD3^+^, CD4^+^, CD8^+^ and CD4^+^/CD8^+^: general anaesthesia versus general anaesthesia plus epidural anaesthesia

Data on the impact of combined general and epidural anaesthesia, compared to general anaesthesia alone, could only be analysed for a period of 72 h following the surgery. The pooled results of two studies [23,24] demonstrated that general plus epidural anaesthesia led to a higher CD4^+^ cell count 72 h after the surgery (SMD 0.59, CI 95% 0.26 to 0.93, I^2^ 0.0%) (Figure 8), while CD3^+^ cells followed a less pronounced reduction (SMD 0.50, CI 95% −0.04 to 1.05, I^2^ 63.11%) (Figure 9). In relation to the CD4^+^/CD8^+^ ratio, the study conducted by Zhou et al. [24] did not observe any significant difference between the two groups 72 h postoperatively. However, Zhao et al. [23] reported that patients who underwent combined general and epidural anaesthesia maintained CD4^+^/CD8^+^ levels 72 h postoperatively similar to preoperative values, whereas patients who received general anaesthesia alone exhibited a decreased ratio. The pooled results showed that the combination of anaesthetic techniques did not significantly increase the CD4^+^/CD8^+^ ratio 72 h after the operation (SMD 0.42, CI 95% −0.21 to 1.05, I^2^ 72.4%) (Figure 10).

#### 3.3.2. Analysis of Secondary Outcomes

a.NLR

Two studies [20,21] examined the impact of propofol or sevoflurane on the NLR. Kim et al. [20] observed that the NLR was significantly lower in the TIVA group, compared to the volatile group. Moreover, Oh Chung-Sik et al. [21] reported that there was no significant difference in the NLR between the two groups. Data were extracted regarding the NLR, immediately after the operation and 24 h postoperatively, while the results indicated that the anaesthesia technique did not have a significant impact on the NLR at either time points (SMD 0.04, CI 95% −0.82 to 0.90, I^2^ 92.71% and SMD −0.12, CI 95% −0.41 to 0.17, I^2^ 36.12%) (Figure 11 and Figure 12, respectively)

b.IL-6

Compared to general anaesthesia alone, the combination of general and epidural anaesthesia led to lower IL-6 levels after surgery, in two [23,24] out of the three studies [22,23,24]. However, these findings could not be confirmed by the pooled estimate of our meta-analysis (SMD 1.56, CI 95% −1.09 to 4.21, I^2^ 98.53%.) (Figure 13). In a prespecified sensitivity analysis, the only study with a high risk of bias [22] was excluded. This particular study was also the only one that included patients with colorectal carcinoma, showing that general epidural anaesthesia was associated with elevated IL-6 levels. These results were also characterized by pronounced heterogeneity, while statistical significance could not be achieved (SMD 2.44, CI 95% −1.10 to 5.97, I^2^ 98.2%) (Figure 14). Two RCTs [22,24] provided data on IL-6 levels 72 h after the operation. The pooled results showed no significant difference between the two groups comparing general and epidural anaesthesia (SMD 1.63, CI 95% −2.16 to 5.42, I^2^ 98.79%) (Figure 15).

c.TNF-a

The pooled results from three studies [22,23,24] investigating the effects of general anaesthesia versus general anaesthesia plus epidural anaesthesia did not reveal any significant difference in the postoperative TNF-a concentration (SMD 0.77, CI 95% −0.66 to 2.20, I^2^ 95.9%) (Figure 16). In a post-hoc sensitivity analysis, where only studies [23,24] that included patients with gastric carcinoma were analysed, the findings remained statistically insignificant (SMD 1.38, CI 95% −0.01 to 2.76, I^2^ 92.6%) (Figure 17). The analysis using data after 72 h also showed no difference between the two methods (SMD 0.15, CI 95% −0.90 to 1.20, I^2^ 90.8%) (Figure 18).

## 4. Discussion

The purpose of this systematic review and meta-analysis was to investigate the impact of different anaesthetic techniques on immune system changes, considering cellular immunity inflammatory cytokines and the leukocyte-to-neutrophil ratio. From the limited number of relevant RCTs incorporated in our analysis, no particular differentiation occurred among TIVA and volatile-based anaesthesia, regarding the primary and secondary endpoints. The combination of general anaesthesia with epidural analgesia exerted a favourable effect on the NK cell count and CD4^+^, while CD3^+^, the CD4^+^/CD8^+^ ratio, NLR IL-6 and TNF-α remained unaffected.

T lymphocytes and NK cells have potential antitumor effects, controlled by several mechanisms in surgical patients. NK cells are related to the antibody-dependent cytotoxicity that mediates both the innate and adaptive immunity [21,25]. Moreover NK cells were previously proved to mediate colorectal cancer progression perioperatively [26]. Recent evidence indicates a direct effect of anaesthesia on the NK cells, which have been identified as major contributors to the immune cancer control as they exhibit potent antitumor and anti-metastatic properties [27,28,29]. Considering that NK cells represent one of the most recent therapeutic targets in oncology, their stimulation during the perioperative period seems to be crucial for cancer prognosis [30].

In the present study, the analysis between TIVA or volatile effects suggested a potential beneficial effect of propofol, after the induction of anaesthesia, as well as 24 h postoperatively, which, however, could not be statistically confirmed. The results should be carefully evaluated since they could not be solidified though sensitivity analysis and due to the fact that great heterogeneity was noted. In addition, the analysis of studies characterized by a low risk of bias agreed with the main results. Notably, our meta-analysis revealed that the combination of general and epidural anaesthesia can significantly increase the NK cell count, an effect being valid up to 72 h postoperatively. Regarding the beneficial effect of propofol, compared with volatile agents, our results agree with an older systematic review and meta-analysis, involving patients with cancer [31], which proposes that inhalational anaesthesia suppresses NK cell activity. Unfortunately, the mechanism by which these effects occur is still not fully understood. Two studies have demonstrated that volatile anaesthesia may result in the reduced activation of NK cells, in response to interferon [32,33]. Additionally, they can lead to the suppression of cytokine release, particularly TNF-α, which ultimately diminishes NK cell activity [32,33,34].

Moreover, the results from two studies on T lymphocyte activity were controversial considering propofol or sevoflurane administration. The beneficial effect of propofol on the CD8^+^ count was attenuated 24 h postoperatively in the study of Chen Yi Jiao et al. [19], while in the study of Oh CS et al. [21], sevoflurane promoted a CD8^+^ count augmentation in the immediate postoperative period and 24 h later. Our pooled analysis revealed that propofol slightly reduced the CD8^+^ count during the immediate postoperative period, in comparison to sevoflurane, and that this reduction was reversed after 24 h, when a slight increase was apparent. However, statistical significance was not achieved.

The pooled results of two studies, comparing general anaesthesia alone or in combination with epidural anaesthesia, revealed a beneficial clinical effect of the combined technique, inducing an elevated or stable CD4^+^ count. However, the combination of anaesthetic techniques did not increase the CD4^+^/CD8^+^ ratio significantly, even 72 h after the surgery. According to the literature, CD3^+^, CD4^+^ and CD8^+^ are categorized as tumour-infiltrating. Tumour-infiltrating lymphocytes (TILs) are the major type of infiltrating immune cells [35,36]. The count of TILs is considered an indicator of the immune response against cancer cells [37]. It is hypothesized that stress during surgery can significantly contribute to immunosuppression and should be recognized as one of the main mechanisms that promote immunosuppression by activating the hypothalamus–pituitary–adrenal cortex axis and leading to the inhibition of T cells, through the release of cortisol and other stress mediators [38]. The combination of general anaesthesia and epidural anaesthesia can reduce these effects, inhibit sympathetic nerve activity and alleviate the cellular immunosuppression caused by surgical stress [39,40].

With regard to the secondary outcomes of our analysis, no profound impact of the anaesthesia regimen or anaesthetic technique on NLR could be demonstrated. Recent studies have proven that NLR, which is a systemic inflammatory response biomarker, can be used as a predictor of survival in colorectal or gastric cancer patients [41]. Although existing data support NLR as an indicator of postoperative and chemotherapy prognosis, there are some concerns about its validity on the basis that NLR is related to the tumour morphology, a factor that limits its reliability and clinical utility [42].

Moreover, recently, great progress has been made in understanding the role of circulating cytokines in cancer prognosis. Published data revealed that high IL-6 levels can be an independent biomarker for a bad prognosis in patients with gastrointestinal cancer, while the results for colorectal cancer were controversial [43]. Our meta-analysis showed no significant difference between general and combined anaesthesia in IL-6 levels, probably due to a pronounced heterogeneity.

Among cytokines related to immunomodulation, TNF-a has been recognized as the most significant proinflammatory cytokine involved in the pathway of carcinogenesis [44]. Recent studies revealed that pro-inflammatory mediators, such as TNF-a, may increase the risk of malignancy [45]. In our analysis, the pooled results from three studies did not show any statistically significant change in the TNF-a concentration between general or combined general plus epidural anaesthesia. The absence of statistical significance could be probably explained by the recent findings that TNF-a variants related to colorectal and gastric cancer are population- and ethnicity-specific [46].

Another aspect to consider is the role of perioperative transfusions, which are routinely conducted during surgical procedures. Allogeneic blood transfusions have been demonstrated to impact various immune functions in recipients, including a reduction in the total number of lymphocytes, decrease in the number of CD4^+^ cells, decrease in the CD4^+^/CD8^+^ T cell ratio and decrease in the absolute number of NK cells [47]. Transfusion-related immunomodulation (TRIM) can potentially result from the various components, within a packed red blood cell (PRBC) transfusion, which includes erythrocytes, transfused leukocytes, platelets and soluble factors originating from these cellular components or already present in the donor plasma.

More specifically, TRIM could be induced through the secretion of Type Th2 immunosuppressive cytokines, as well as the release of Type Th1 pro-inflammatory cytokines released by WBCs during the storage of blood products [48,49]. The supernatant of PRBCs is able to induce the activation of regulatory T cells (Tregs), thereby suppressing the Th1 immune response [50]. In a recent study, the levels of mRNA for a group of interconnected cytokines and related transcription factors were examined in patients who underwent the transfusion of allogeneic blood [51]. The study demonstrated that patients who received a perioperative blood transfusion following major gastrointestinal surgery exhibited a gene expression pattern indicative of heightened immunosuppression compared to a cohort that did not undergo a blood transfusion [51]. Data regarding transfusions within the studies included in this meta-analysis were not available. These effects can influence the immunomodulation observed from different anaesthesia techniques, potentially serving as a confounding factor.

Additionally, it is important to consider the variation in analgesia protocols across the studies during surgical procedures. The diverse forms of analgesia exert a significant influence on perioperative immunomodulation. Notably, opioids consist of the most frequently administered analgesics, extensively utilized in the context of cancer surgeries. It has been established that opioids can suppress both cellular and humoral immunity, thereby promoting a state of immune suppression [52,53]. Their primary effect is mediated through the μ-opioid receptor (MOR gene). Morphine, for instance, exerts its effects by suppressing NK cell activity, hindering T cell differentiation, catalysing lymphocyte apoptosis and diminishing the expression of toll-like receptor 4 (TLR4) on macrophages [54,55,56]. Opioids also interact with various cytokines, such as IL-1, IL-4, IL-6 and TNF-a, in distinct ways [57]. For example, while fentanyl enhances IL-4 production by T cells, morphine seems to inhibit it [58]. Moreover, in a study involving mice and small-cell lung carcinoma, the elimination of the MOR gene led to a reduction in tumour growth and metastases [59].

Non-steroidal anti-inflammatory drugs (NSAIDS) are another category of analgesics that should be taken into consideration. Enzymes that regulate the production of prostaglandins and cyclooxygenase are often overexpressed in malignancies [60]. The COX-2–PGE2 signalling pathway could potentially aid tumours in avoiding immune surveillance. This could occur through mechanisms, such as promoting the aggregation of immune cells around tumours, reducing the activity of antigen-presenting cells, shifting immune responses from Th1 to Th2 and Th17 or inhibiting the functions of CD8^+^ cytotoxic T cells and NK cells, all of which contribute to facilitating tumour immune escape. Consequently, medications that target the production of PGE2 and COX-2 are linked to potential immunomodulatory effects [61]. A recent meta-analysis has demonstrated that COX-2 overexpression is linked to a poorer prognosis, characterizing it as an independent prognostic factor for patients with osteosarcoma [62]. Despite the available evidence, the impact of NSAIDs remains somewhat unclear. These findings underscore the complex interplay between analgesia and immunomodulation. The profound variation in the analgesic protocols applied for perioperative pain management serves also as a confounding factor that could have influenced the observed results and needs to be separately explored.

We acknowledge that the present study has certain limitations, which should be taken into consideration. The analysis was based on a limited number of studies, and significant heterogeneity was observed among them. In addition, the included studies varied also in terms of methodology, population types of surgery disease stage and the anaesthetic techniques used. To address the above difficulties, an available-case analysis was conducted, while the last observation was carried forward for continuous data. Albeit these statistical approaches are well-documented and yield reliable results, caution should be exercised when interpreting the findings. Moreover, it is worth mentioning that the main results of this study may be susceptible to statistical bias as the scarcity of data prevented us from conducting the predetermined sensitive analysis.

In the future, the results of our analysis can be combined with the impact of anaesthetic agents and practices during operations for other malignancies, in order to finally reveal if there is an ideal technique for cancer surgical patients. However, the only way to safely conclude on certain anaesthetic plans used during the surgical excision of a tumour is through an analysis of existent high-quality randomized controlled trials.

## 5. Conclusions

Given the unique role of immunomodulation in cancer progression and the distinctive need to elucidate possible modifiers of this process, the supplementation of general anaesthesia with epidural analgesia might exert a favourable effect on the NK cell count and CD4^+^ levels in a subset of patients subjected to gastric or colon surgery, while TNF-α and IL-6 levels, as well as CD3^+^, the NLR and the CD4^+^/CD8^+^ ratio remained unaffected. Of note, the clear-cut role of TIVA or volatile-based anaesthesia in this setting could not be ascertained. However, due to the heterogeneity and inconsistent results of the limited available evidence on this topic, no definite conclusions could be drawn. Thus, our study serves as a call for future well-designed comparative trials with homogeneous methodologies to elucidate the potential immunomodulatory role of the applied anaesthetic practice in these subsets of cancer patients.

## Figures and Tables

**Figure 1 jcm-12-06027-f001:**
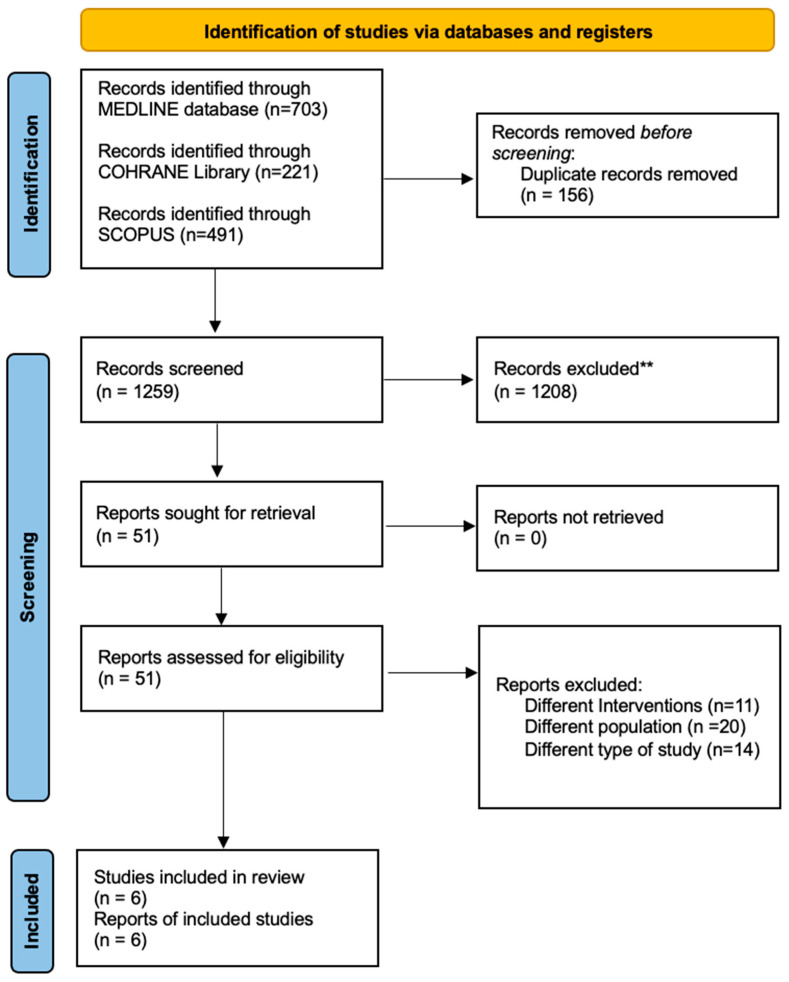
PRISMA 2020 Flow-diagram.

**Figure 2 jcm-12-06027-f002:**
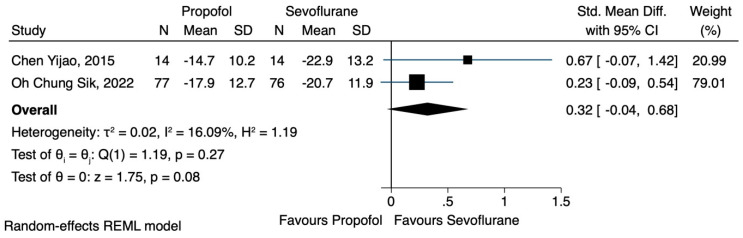
NK cell count immediately after the operation.

**Figure 3 jcm-12-06027-f003:**
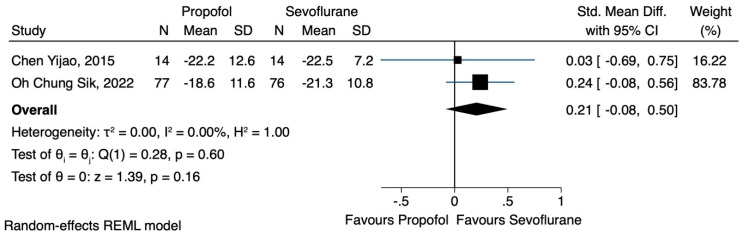
NK cell count 24 h after the operation.

**Figure 4 jcm-12-06027-f004:**
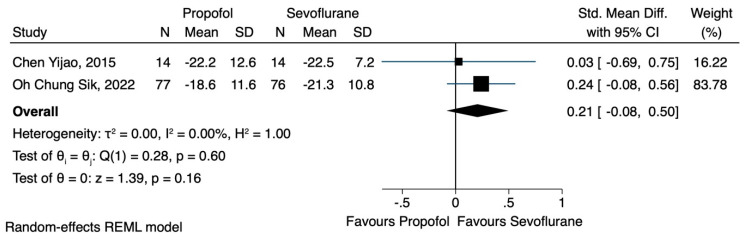
NK cell count after the induction of anaesthesia.

**Figure 5 jcm-12-06027-f005:**
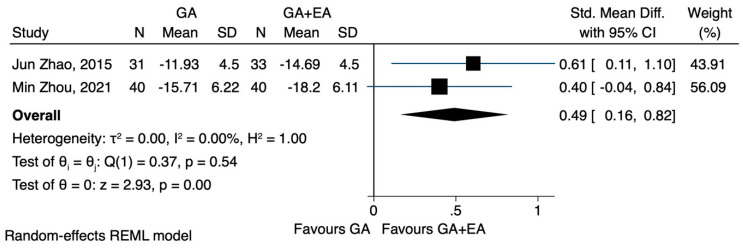
NK cell count 72 h after the operation.

**Figure 6 jcm-12-06027-f006:**
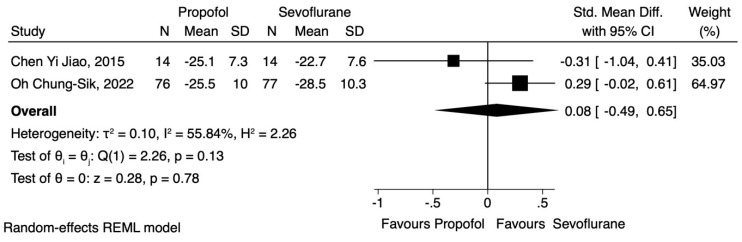
CD8^+^ lymphocyte count immediately after the operation.

**Figure 7 jcm-12-06027-f007:**
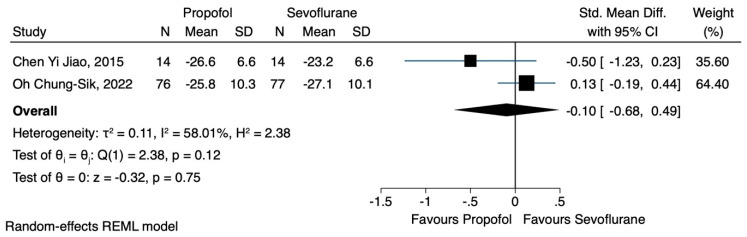
CD8^+^ lymphocyte count 24 h after the operation.

**Figure 8 jcm-12-06027-f008:**
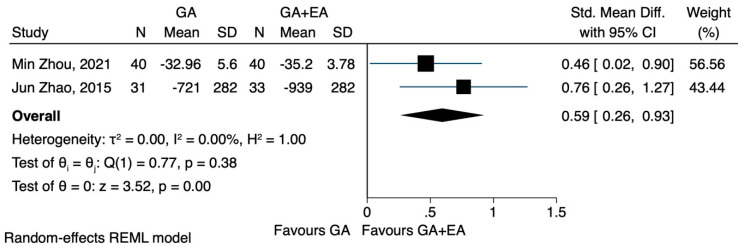
CD4^+^ lymphocyte count 72 h after the operation, GA + EA vs. GA.

**Figure 9 jcm-12-06027-f009:**
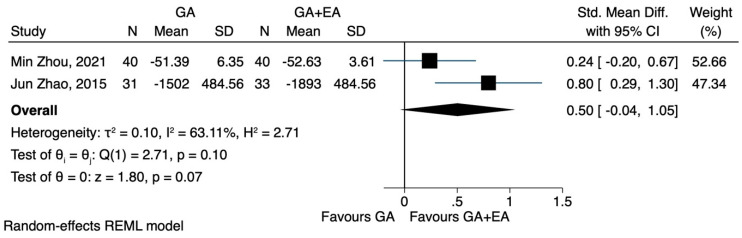
CD3^+^ lymphocyte count 72 h after the operation, GA + EA vs. GA.

**Figure 10 jcm-12-06027-f010:**
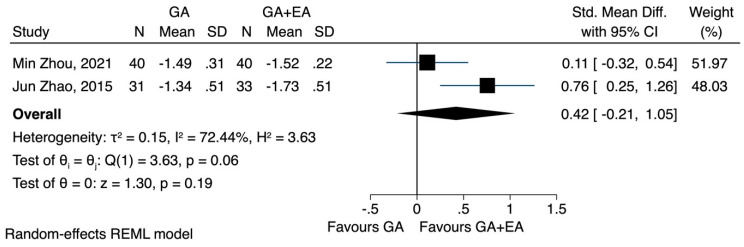
CD3^+^/CD8^+^ lymphocyte count 72 h after the operation, GA + EA vs. GA.

**Figure 11 jcm-12-06027-f011:**
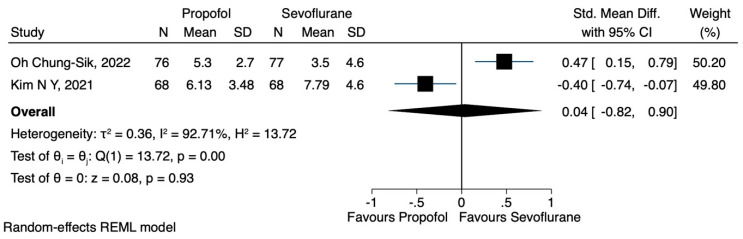
NLR after the operation, propofol vs. sevoflurane.

**Figure 12 jcm-12-06027-f012:**
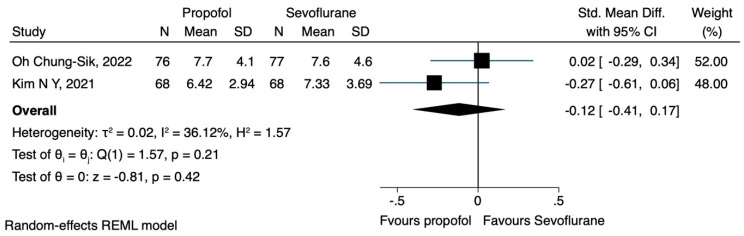
NLR 24 h postoperatively, propofol vs. sevoflurane.

**Figure 13 jcm-12-06027-f013:**
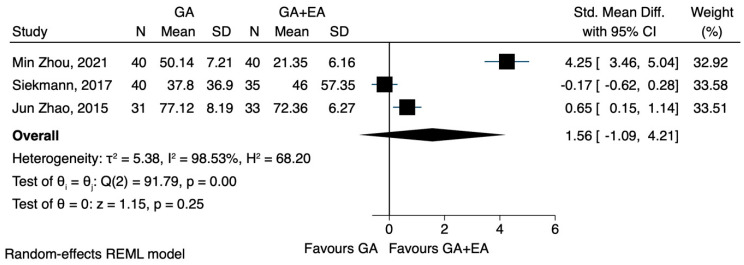
IL-6 postoperatively.

**Figure 14 jcm-12-06027-f014:**
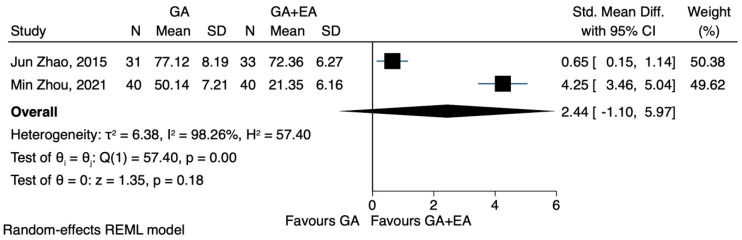
IL-6 postoperatively, sensitivity analysis.

**Figure 15 jcm-12-06027-f015:**
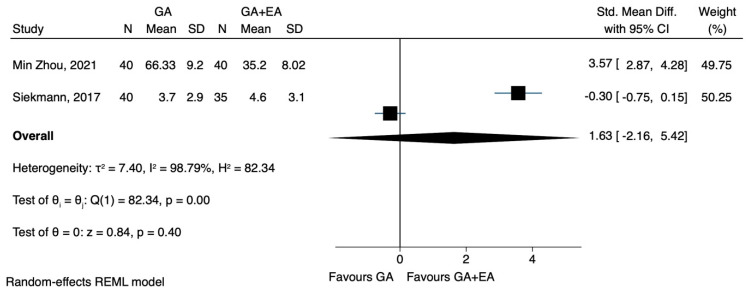
IL-6 72 h postoperatively.

**Figure 16 jcm-12-06027-f016:**
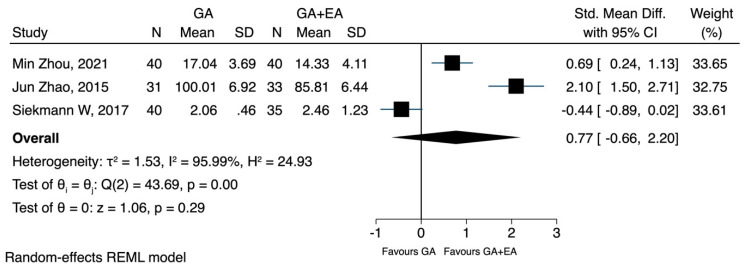
TNF-a postoperatively.

**Figure 17 jcm-12-06027-f017:**
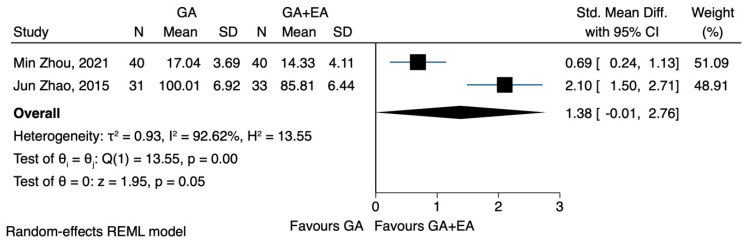
TNF-a, sensitivity analysis.

**Figure 18 jcm-12-06027-f018:**
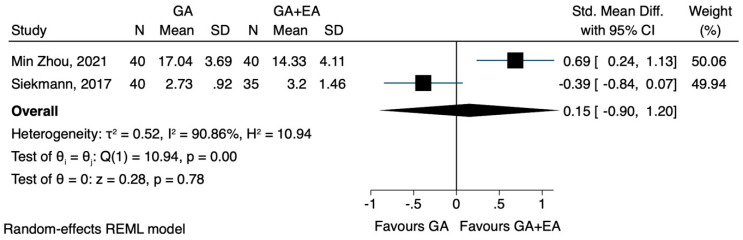
TNF-a 72 h postoperatively.

**Table 1 jcm-12-06027-t001:** Basic characteristics of included studies.

First Author, Year, Country	Number of Participants (Male/Total)	Cancer Subtype	TNM Classification Group 1	TNM Classification Group 2	Type of Surgery	ASA	Age of Participants
Chen, Y. [19], 2015, China	30 Propofol group: 7/14 Sevoflurane group: 9/14	Colorectal cancer	Sevoflurane T1-3 N0-2 M0	Propofol T1-3 N0-2 M0	Laparoscopic radical surgery for colorectal cancer	Sevoflurare group ASA I: 1 ASA II: 13 Propofol group ASA I:0 ASA II: 14	Sevoflurane group: 56 ± 12 Propofol group: 61 ± 12
Zhou, M. [24], 2021, China	80 General anaesthesia group: 24/40 General plus EA: 28/40	Gastric cancer	N/A	N/A	Laparoscopic radical gastrectomy	N/A	General anaesthesia group: 66.3 ± 6.9 General plus EA: 65.4 ± 7.2
Oh, C.S. [21], 2022, Korea	153 Propofol group: 39/76 Sevoflurane group: 40/77	Colorectal cancer	N/A	N/A	Propofol group: hemicolectomy, 16; transverse colectomy, 4; low anterior resection, 35; anterior resection, 12; abdominoperineal, 9 Sevoflurane group: hemicolectomy, 14; transverse colectomy 1; low anterior resection, 41; anterior resection, 14; abdominoperineal, 7;	Sevoflurane group ASA I: 29 ASA II: 37 ASA III: 11 Propofol group ASA I: 33 ASA II: 35 ASA III: 8	Sevoflurane group 64.4 ± 11.3 Propofol group: 62.2 ± 9.8
Siekmann, W. [22], 2017, Sweden	96 General anaesthesia group: 24/43 General plus EA: 30/37	Colorectal cancer	General anaesthesia group TNM 1:1, TNM 2:4 TNM 3:33 TNM 4: 3	General anaesthesia plus EA: TNM 1:2 TNM 2:5 TNM 3:23 TNM 4:5	right hemicolectomy, resection of the transverse colon, left hemicolectomy or sigmoid resection were employed, as appropriate. For rectal cancer, either anterior resection, abdomino-perineal resection or Hartmanns resection	I-III	General anaesthesia group: 68 ± 9.6 General anaesthesia plus EA: 69 ± 7.1
Zhao, J. [23], 2015, China	64 General anaesthesia group: 14/31 General plus EA: 15/33	Gastric cancer	N/A	N/A	Radical resection of antral cancer	N/A	General anaesthesia group: 52.8 ± 3.4 General anaesthesia plus EA: 53.4 ± 2.9
Kim, N.Y. [20], 2021 Korea	136 Propofol: 68 Sevoflurane: 68	Gastric cancer	Sevoflurane group TNM 1: 41 TNM 2: 14 TNM 3: 7 TNM 4: 3	Propofol group TNM 1: 47 TNM 2: 12 TNM 3: 4 TNM 4: 5	Propofol: subtotal gastrectomy, 58; total gastrectomy, 7; proximal subtotal gastrectomy, 2; Sevoflurane: subtotal gastrectomy, 51; total gastrectomy, 12; proximal subtotal gastrectomy, 2	Propofol: ASA I: 8 ASA II: 45 ASA III: 14 Sevoflurane ASA I: 9 ASA II: 39 ASA III: 17	Propofol group: 61.5 ± 9 Sevoflurane group: 64.6 ± 10.4

Values are presented as the mean  ±  SD; ASA: American Society of Anesthesiologists; TNM: tumour-node-metastasis; EA: epidural anaesthesia.

**Table 2 jcm-12-06027-t002:** Anaesthesia characteristics.

First Author, Year, Country	GA Maintenance	Epidural Catheter Characteristics	Duration of the Epidural Catheter	Epidural Medications Intraoperatively	Intraoperative Opioids	Postoperative Analgesia
Chen, Y. [19], 2015, China	Propofol Sevoflurane	N/A	N/A	N/A	Sulfentanil 0.3–0.5 μg/(kg·min)	NSAID PCA
Zhou, M. [24], 2021, China	Propofol	The epidural catheter was placed between T8 and T9, with a depth of 4 cm	N/A	N/A	Fentanyl 2 μg/kg	N/A
Oh, C.S. [21], 2022, Korea	Propofol Sevoflurane	N/A	N/A	N/A	Remifentanil at 5 ng mL^−1^	Fentanyl PCA
Siekmann, W. [22], 2017, Sweden	Propofol	The epidural catheter was placed at the T10-12, preoperatively	Ca 72 H	Bupivacaine 2 mg/mL, adrenaline 5 μg/mL and fentanyl 1 μg/mL or bupivacaine 5 mg/mL	In propofol group, fentanyl was given as needed.	General anaesthesia plus EA: epidural anaesthesia with fentanyl/sulfentanyl. After epidural, postoperative analgesia was stopped and patients received NSAIDs General anaesthesia: i.v morphine All patients in both groups received paracetamol.
Zhao, J. [23], 2015, China	Isoflurane	The epidural catheter was placed at thoracic vertebrae (T7-8)	N/A	Lidocaine at 20 g/L	Fentanyl 0.02 μg/(kg·min)	N/A
Kim, N.Y. [20], 2021 Korea	Propofol Sevoflurane	N/A	N/A	N/A	Remifentanil Sevoflurane group: 940 ± 330 mg Propofol group: 1294 ± 464 mg	Fentanyl PCA

i.v: intravenous; kg: kilogram; mg: microgram; NSAID: Nonsteroidal anti-inflammatory drugs; T: Thorax; EA: epidural anaesthesia; N/A: not applicable; GA: general anaesthesia; PCA: patient control analgesia.

**Table 3 jcm-12-06027-t003:** Risk of bias assessment for primary outcome with Cohrane RoB tool 2.0.

First Author, Year, Country	Randomization Process	Deviations from Intended Interventions	Missing Outcome Data	Measurement of the Outcome	Selection of the Reported Result	Overall Bias
Chen, Y. [19], 2015, China	Some concerns	Some concerns	Low	Low	Some concerns	Some concerns
Zhou, M. [24], 2021, China	Some concerns	Some concerns	Low	Low	Some concerns	Some concerns
Oh, C.S. [21], 2022, Korea	Low	Low	Low	Low	Low	Low
Siekmann, W. [22], 2017, Sweden	High	Some concerns	Low	Low	Some concerns	High
Zhao, J. [23], 2015, China	Some concerns	Some concerns	Low	Low	Some concerns	Some concerns
Kim, N.Y. [20], 2021, Korea	Some concerns	Low	Low	Low	Low	Low

RoB: Risk of Bias.

## Data Availability

Not applicable.

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
