# Peer review of "The Immunomodulatory Effect of Various Anaesthetic Practices in Patients Undergoing Gastric or Colon Cancer Surgery: A Systematic Review and Meta-Analysis of Randomized Clinical Trials"

_jcm, 2023, doi:10.3390/jcm12186027_

Round 1

Reviewer 1 Report

My suggestions to improve the article

- influence of perioperative transfusions and their role in immunosuppression

- influence of analgesia (which is different from an efficacy point of view between the analysed ways of anaesthesia)  on immunosupression

Author Response

Reviewer No 1:

- influence of perioperative transfusions and their role in immunosuppression

Suggestion was followed. The influence of perioperative transfusion and its role in immunosuppression were added in discussion section (line 480 to 506). The added text is the following:

Another aspect to consider is the role of perioperative transfusions, which are routinely conducted during surgical procedure. Αllogeneic blood transfusions have been demonstrated to impact various immune functions in recipients, including: reduction in the total number of lymphocytes; decrease in the number of CD4+ cells, decrease in the CD4+/CD8+ T-cell ratio and decrease in the absolute number of NK cells(47). Transfusion-related immunomodulation (TRIM) can potentially result from the various components, within a packed red blood cell (PRBC) transfusion, which includes erythrocytes, transfused leukocytes, platelets, and soluble factors originating from these cellular components or already present in the donor plasma.

More specifically, TRIM could be induced through the secretion of Type Th2 immunosuppressive cytokines as well as the release of Type Th1 pro-inflammatory cytokines released by WBCs during storage of blood products(48, 49). The supernatant of PRBCs is able to induce the activation of regulatory T-cells (Tregs), thereby suppressing the Th1 immune response(50). In a recent study, the levels of mRNA for a group of interconnected cytokines and related transcription factors were examined in patients who underwent transfusion of allogeneic blood(51). The study demonstrated that patients, who received a perioperative blood transfusion following major gastrointestinal surgery exhibited a gene expression pattern indicative of heightened immunosuppression compared to a cohort that did not undergo a blood transfusion(51). Data regarding transfusions within the studies included in this meta-analysis were not available. These effects can influence the immunomodulation observed from different anaesthesia techniques, potentially serving as a confounding factor.

- influence of analgesia (which is different from an efficacy point of view between the analysed ways of anaesthesia)  on immunosupression

Suggestion was followed. The influence of different analgesics and their role in immunosuppression were added in discussion section (line 507 to 543). The added text is the following:

Additionally, it is important to consider is the variation in analgesia protocols across the studies during surgical procedures. The diverse forms of analgesia exert a significant influence on perioperative immunomodulation. Notably, opioids consist the most frequently administered analgesics, extensively utilized in the context of cancer surgeries. It has been established that opioids can suppress both cellular and humoral immunity, thereby promoting a state of immune suppression(52, 53). Their primary effect is mediated through the μ-opioid receptor (MOR gene). Morphine, for instance, exerts its effects by suppressing NK cell activity, hindering T cell differentiation, catalysing lymphocyte apoptosis, and diminishing the expression of toll-like receptor 4 (TLR4) on macrophages(54-56). Opioids also interact with various cytokines such as IL-1, IL-4, IL-6, and TNF-a in distinct ways(57). For example, while fentanyl enhances IL-4 production by T-cells, morphine seems to inhibit it(58). Moreover, in a study involving mice and small-cell lung carcinoma, the elimination of the MOR gene led to a reduction in tumor growth and metastases(59).

Non steroidal anti-inflammatory drugs (NSAIDS) is another category of analgesics that should be taken into consideration. Enzymes that regulate the production of prostaglandins and cyclooxygenase are often overexpressed in malignancies(60). The COX-2-PGE2 signaling pathway could potentially aid tumors in avoiding immune surveillance. This could occur through mechanisms, such as promoting the aggregation of immune cells around tumors, reducing the activity of antigen-presenting cells shifting immune responses from Th1 to Th2 and Th17, or inhibiting the functions of CD8+ cytotoxic T-cells and NK cells, all of which contribute to facilitating tumor immune escape. Consequently, medications that target the production of PGE2 and COX-2 are linked to potential immunomodulatory effects(61). A recent meta-analysis has demonstrated that COX-2 overexpression is linked to poorer prognosis, characterizing it as an independent prognostic factor for patients with osteosarcoma(62). Despite the available evidence, the impact of NSAIDs remains somewhat unclear. These findings underscore the complex interplay between analgesia and immunomodulation. The profound variation in the analgesic protocols applied for perioperative pain management serves also as a confounding factor, that could have influenced the observed results and needs to be separately explored.

Reviewer 2 Report

Dear authors,

please highlight the novelty of your work in the introduction and abstract sections. Kindly highlight inclusion /exclusion criteria in depth. what would be your future work perspectives, kindly share your view and approaches. 

Author Response

-please highlight the novelty of your work in the introduction and abstract sections.

Suggestion was followed. The novelty of our meta-analysis was added in abstract (line 26 to 30). The added text is the following:

There is no meta-analysis investigating anaesthesia’s impact on immune responses in gastric and colorectal cancer surgery. Anaesthesia is a key perioperative factor, yet its significance in this area hasn't been thoroughly investigated. The clinical question of how anaesthetic technique choice affects the immune system and prognosis remains unresolved.

as well as the introduction section (line 104 to 1110. The added text is the following:

The novelty of the present study lies in the fact that, this is the first systematic review and meta-analysis focusing on the effect of various anaesthesia techniques on the immune system of patients undergoing surgery for gastric or colorectal cancer. Anaesthesia stands as one of the most alterable perioperative factors, and thus far, its importance in this field has not been adequate explored. Clinically, the pivotal question of whether the choice of anesthetic affects the immune system and, consequently, the prognosis remains unanswered.

Kindly highlight inclusion /exclusion criteria in depth.

Suggestion was followed, the inclusion and exclusion criteria were written in more details (line 130 to 137). The added text is the following:

Studies that did not utilize any of the aforementioned anaesthetic technique combinations were excluded from this meta-analysis. The inclusion criteria for participants were limited to adults only. No limitations were imposed on the utilized analgesia protocols. Moreover, no exclusion criteria were established regarding to the type of surgical procedure (endoscopic, open, laparoscopic, robotic), disease stage and comorbidities. Additionally, the duration of hospitalization was not taken into consideration.

what would be your future work perspectives, kindly share your view and approaches. 

Suggestion was followed. The following text was added (line 556 to 561).

In the future, the results of our analysis can be combined with the impact of anaesthetic agents and practices during operation for other malignancies, in order to finally reveal if there is an ideal technique for cancer surgical patients. However, the only way to safely conclude in certain anaesthetic plans used during a surgical excision of a tumor, is through analysis of existent high quality randomized controlled trials.

Reviewer 3 Report

In this review paper, the authors discuss the findings from six separate clinical trials that examined how various anesthetic approaches impact the immune system, particularly the ratios of immune cells, in individuals with gastric and colon tumors. It's worth noting that the authors acknowledge a crucial limitation: the relatively small number of studies analyzed, which influences the outcomes and overall conclusions. Additionally, the manuscript would greatly benefit from thorough proofreading to improve its clarity, flow, and overall ease of reading.

The manuscript needs comprehensive editing for the English language.

Author Response

Additionally, the manuscript would greatly benefit from thorough proofreading to improve its clarity, flow, and overall ease of reading.

The manuscript needs comprehensive editing for the English language.

Suggestion was followed. The whole manuscript has been edited and thoroughly corrected (all the changes are highlighted according to the Editor’s request).

Reviewer 4 Report

Gastric and colorectal cancers are a growing health concern worldwide. This study documents better prognosis for patients operated for gastric and colorectal cancers in combination of epidural analgesia and general anaesthesia. A positive effect on NK-cells count and CD4+ cells count was observed, but a mechanism of improved immune response remains uncler.

This is article is an interesting, properly designed and well written meta-analysis of up-to-date literature drawing attention to a poorly known phenomenon in treatment of cancer patients. I recommend publishing this paper for the interest of readers.

Author Response

Gastric and colorectal cancers are a growing health concern worldwide. This study documents better prognosis for patients operated for gastric and colorectal cancers in combination of epidural analgesia and general anaesthesia. A positive effect on NK-cells count and CD4+ cells count was observed, but a mechanism of improved immune response remains uncler.

This is article is an interesting, properly designed and well written meta-analysis of up-to-date literature drawing attention to a poorly known phenomenon in treatment of cancer patients. I recommend publishing this paper for the interest of readers.

Thank you very much for your comments. We have made an attempt to clarify the opioid-immune system interaction in lines 507 to 523. The added text is the following:

Additionally, it is important to consider is the variation in analgesia protocols across the studies during surgical procedures. The diverse forms of analgesia exert a significant influence on perioperative immunomodulation. Notably, opioids consist the most frequently administered analgesics, extensively utilized in the context of cancer surgeries. It has been established that opioids can suppress both cellular and humoral immunity, thereby promoting a state of immune suppression(52, 53). Their primary effect is mediated through the μ-opioid receptor (MOR gene). Morphine, for instance, exerts its effects by suppressing NK cell activity, hindering T cell differentiation, catalysing lymphocyte apoptosis, and diminishing the expression of toll-like receptor 4 (TLR4) on macrophages(54-56). Opioids also interact with various cytokines such as IL-1, IL-4, IL-6, and TNF-a in distinct ways(57). For example, while fentanyl enhances IL-4 production by T-cells, morphine seems to inhibit it(58). Moreover, in a study involving mice and small-cell lung carcinoma, the elimination of the MOR gene led to a reduction in tumor growth and metastases(59).